# Energy Consumption Minimization with SNR Constraint for Wireless Powered Communication Networks

**DOI:** 10.3390/s24175535

**Published:** 2024-08-27

**Authors:** Kuei-Ping Shih, Yu-Sheng Tsai, Yen-Da Chen, San-Yuan Wang

**Affiliations:** 1Department of Computer Science and Information Engineering, Tamkang University, New Taipei City 251301, Taiwan; 2Department of Computer Information and Network Engineering, Lunghwa University of Science and Technology, Taoyuan 333326, Taiwan; ydchen@mail.lhu.edu.tw; 3Department of Information Engineering, I-Shou University, Kaohsiung 811022, Taiwan; sywang@isu.edu.tw

**Keywords:** Harvest Then Transmit (HTT) protocol, Simultaneous Wireless Information and Power Transfer (SWIPT), SNR constraint, wireless energy transfer, Wireless Powered Communication Networks (WPCNs)

## Abstract

The article addresses the energy consumption minimization problem in wireless powered communication networks (WPCNs) and proposes a time allocation scheme, named DaTA, which is based on the Different Target Simultaneous Wireless Information and Power Transfer (DT-SWIPT) scheme such that the wireless station can share the remaining energy after transmission to the Hybrid Access Point (HAP) to those who have not transmitted to the HAP to minimize the energy consumption of the WPCN. In addition to proposing a new frame structure, the article also considers the Signal-to-Noise (SNR) constraint to guarantee that the HAP can successfully receive data from wireless stations. In the article, the problem of minimization of energy consumption is formulated as a nonlinear programming model. We employ the SQP (Sequential Quadratic Programming) algorithm to figure out the optimal solution. Moreover, a heuristic method is proposed as well to obtain a near-optimal solution in a shorter time. The simulation results showed that the proposed scheme outperforms the related work in terms of energy consumption and energy efficiency.

## 1. Introduction

In wireless networks, including but not limited to wireless sensor networks and wireless body area networks, it is typically necessary for a wireless station to be equipped with at least one battery for power. However, it is widely recognized that batteries pose significant environmental pollution issues and should ideally be recycled. However, the retrieval of these batteries is often challenging because of the substantial costs associated with their widespread deployment and inaccessible locations. Moreover, battery-related issues consistently present operational challenges within the network. While battery replacement or recharging may offer a potential solution to prolong the network’s lifespan, such measures are often impractical in many real-world scenarios.

Recently, a novel technology known as Wireless Energy Transfer (WET) has been explored, which employs radio frequency (RF) to transmit energy to wireless stations [1,2,3]. With the evolution of WET and antenna capabilities, a new paradigm of networks, termed Wireless Powered Communication Networks (WPCNs), has emerged in the current communication networks. In WPCNs, batteries are substituted by capacitors to alleviate the burden induced by battery usage. In conventional communication networks, wireless stations are powered by batteries. However, in WPCNs, wireless stations gain energy through WET technology. Thus, the wireless station in WPCNs does not require a battery but a capacitor for energy storage. With the attributes of WPCNs, wireless stations can circumvent the pollution and consumption issues instigated by the large-scale usage of batteries. With the escalating consciousness of green communication [4], the issues in WPCNs have become pivotal and significant. WPCNs can be implemented in a broad spectrum of applications, such as Wireless Sensor Networks (WSNs), Internet of Things (IoT), and Wireless Body Area Networks (WBANs), among others. Typical scenarios of WPCNs encompass smart homes, remote surgery, hazardous areas like volcanoes, and so forth.

Generally, a WPCN is composed of a Power Provider (PP), an Access Point (AP), and several wireless stations. The PP is responsible for providing energy by WET technology to wireless stations, while AP is responsible for receiving data from wireless stations. A wireless station needs to be charged from the PP until the energy harvested from the PP is enough for the wireless station to transmit data to the AP. Then the wireless station transmits the data to the AP. The transmission protocol is called *Harvest-Then-Transmit* (HTT) protocol, which is mainly divided into two phases: *ET* (Energy Transfer) and *IT* (Information Transmission) phases. The phase that the PP transfers energy to the wireless stations is the ET phase and the phase that wireless stations transmit data to the AP is the IT phase. The device playing the two roles, PP and AP, simultaneously is called an HAP (Hybrid AP).

Recently, many researches focus on the WPCNs, such as [5,6,7,8,9,10,11,12,13,14,15,16,17]. Different network models and scenarios of WPCNs have been investigated. Among them, most studies adopt the HTT protocol and focus on how to allocate transmission time of wireless stations and HAP or allocate transmission power of the HAP to address the problems. In general, the problems can be divided into 3 categories.

**STM** problem: sum-throughput maximization problem [6,9,13,14,15],**EEM** problem: energy efficiency maximization problem [5,10,12], and**ECm** problem: energy consumption minimization problem [7,11,16,17].

The STM problem aims to maximize the sum-throughput, which is the total throughput of all wireless stations during the HTT period. The EEM problem aims at maximizing the sum of the ratio of throughput to the energy consumption of each wireless station. However, the ECm problem aims to minimize the energy consumption of the network while meeting the throughput requirements of each wireless station when the HAP has energy limitations.

In the STM problem, the authors in [13,14] considered the PP and the AP in the network and introduced backscatter technology so that wireless stations can receive energy signals from the PP during the ET phase, modulate them, and reflect to the AP. The researchers in [13,14] considered how to maximize the sum-throughput of the network by allocating the backscatter time and transmission time of each wireless station. In addition, the authors in [9,15] considered that the HAP not only transfers energy, but also transmits information to wireless stations. Therefore, the goal of [9,15] is to allocate the ET and IT time of the HAP and the transmission time of each wireless station to maximize the sum-throughput of the network. The authors in [6] considered a WPCN-based WSN. Through centralized management, all wireless sensors are scheduled by the HAP. The article [6] considered a dual-hop network model. Wireless sensors are divided into groups. Each group will select a wireless sensor as the group header. The group header is responsible for summarizing the information of all wireless sensors in the same group. Moreover, the researchers in [6] introduced Simultaneous Wireless Information and Power Transfer (SWIPT) technology to allow wireless sensors to transmit information and transfer excess energy to the group header to relieve the group header’s burden on energy demand to summarize and transmit information to the HAP.

The STM problem is to consider how to maximize the sum-throughput of the network. However, only considering the sum-throughput of the network will cause excessive energy consumption and result in low energy efficiency. Therefore, two different types of research focusing on maximizing energy efficiency or minimizing energy consumption are proposed instead of only considering sum-throughput maximization. The research on maximizing energy efficiency is termed the EEM problem [5,10,12] and that on minimizing energy consumption is termed the ECm problem [7,11,16,17]. The EEM problem aims at maximizing the sum of the ratio of throughput to the energy consumption of each wireless station. On the other hand, the ECm problem aims to minimize the energy consumption of the network while meeting the throughput requirements of each wireless station when the HAP has energy limitations.

In the EEM problem, the paper [12] considered a network model with a PP and an AP. Wireless stations adopt the frequency division multiple access (FDMA) protocol, which allows wireless stations to transmit information to the AP at the same time. Therefore, the authors in [12] achieve maximum energy efficiency by controlling the transmission power of the PP. The network model considered by [5,10] contains an HAP. The wireless stations use the TDMA (Time Division Multiple Access) protocol, which allows the wireless stations to transmit information to the HAP sequentially and achieve the goal through time allocation and power control of the HAP.

In the ECm problem, the researchers in [11] considered a centralized wireless sensor network with an energy-limited HAP. In this network model, wireless sensors are divided into two groups: a group of collaborators that are responsible for transmitting data to the HAP and a group of enjoyers that only receive energy from the HAP. In addition to the throughput requirements of the collaborators, the authors in [11] also considered the energy requirements of the enjoyers. The researchers in [17] took the full-duplex HAP into account, in which the HAP will transfer energy continuously, and wireless stations harvest energy not only in the ET phase but also in the IT phase until the wireless stations start to transmit information to the HAP. The authors in [16] focused on a wireless body area network and follow the network model of [11]. The authors in [16] introduce SWIPT technology in the ET phase so that the HAP can transmit energy and information to wireless stations simultaneously. The researchers in [7] considered a large-scale WPCN which contains several PPs and wireless stations. The wireless stations will be divided into several groups. Each group will have a sub-sink responsible for collecting information from its members. Since the paper [7] considered multiple PPs, wireless stations used the FDMA protocol to transmit information to the AP at the same time. The scheme proposed in [7] also controls the power of each PP to minimize energy consumption.

However, in wireless communication, the SNR (Signal-to-Noise Ratio) value of a received signal should be greater than a certain value to be recognized as legitimate information, which is also known as the SNR constraint [18]. Nevertheless, all the above-mentioned studies did not take the SNR constraint into account. Therefore, this paper will consider the SNR constraint to ensure that the SNR of the received signal from the HAP is greater than a certain threshold so that the HAP can successfully receive the information from the wireless stations. Moreover, with more and more WPCN applications, mobile phones, and even smart bracelets can act as HAP. Therefore, in most cases, the energy of the HAP is limited. Due to the energy limitation of the HAP, when the wireless station meets the throughput demand, saving the energy of the HAP becomes more important than increasing the energy efficiency of the wireless stations. Consequently, the goal of this paper is to minimize the energy consumption of the network under the condition that the wireless stations meet their throughput demands.

Different from the SWIPT scheme [6], the paper proposes a new scheme, called Different Target SWIPT (DT-SWIPT), which relieves the constraint that the receiver of the information transfer should be the same station of the power transfer. In DT-SWIPT, a wireless station finished information transmission can still transfer energy to other stations to share the excess energy. That is, DT-SWIPT is not limited that the information receiver and the power receiver are the same. Based on the above, we formulate a minimization problem based on the DT-SWIPT scheme into a non-linear programming model. The model is termed the DT-SWIPT assisted Time Allocation (DaTA) for the ECm problem. We adopt a Sequential Quadratic Programming (SQP) [19] algorithm to solve the problem. Since the optimization problem may take a long time, from $O(n^{3})$ [11] to $O(2^{n})$ depending on the problem category, to solve, the paper also proposes a heuristic method, called DaTA-H, to obtain an approximate solution in a reasonable time.

The contributions of the paper are summarized as follows.

Different from the general SWIPT, in this paper, we propose a new DT-SWIPT mechanism, which allows wireless stations that finish information transmission to share their excess energy to other wireless stations that have not yet transmitted, thereby reducing energy consumption in the network and overcoming the doubly near-far problem.Ensure that the SNR value of the received signal of the HAP can meet the SNR threshold so that the HAP can decode the signals from wireless stations.The optimization model, named DaTA, is proposed to obtain the best solution for the ECm problem. In addition, the DaTA-H scheme is also proposed so that an approximate solution can be obtained in a reasonable time.Two scenarios, special scenarios and general scenarios are conducted in the simulation to verify the advantages of the proposed schemes. We compare the DaTA and the DaTA-H schemes proposed in this paper against the STM, EEM, and ECm models of the related works. From the simulation results, it can be concluded that the DaTA scheme has better performance in the energy consumption and the energy efficiency in both general and special scenarios. In special scenarios, when the SNR threshold reaches a certain value, the energy consumption and the energy efficiency of the DaTA-H scheme are also better than those of the related works.

The remaining of the paper is structured as follows. Section 2 will introduce the network model, the DT-SWIPT operation method, and the frame structure in detail. Section 3 will introduce the optimization model established in this paper. Section 4 describes the heuristic approach, the DaTA-H scheme. Performance evaluations are shown in Section 5. Finally, Section 6 concludes the summary and contribution of this paper and points out future works.

## 2. Preliminaries

In this section, Section 2.1 introduces the network model of this paper. Section 2.2 introduces the differences between the DT-SWIPT scheme proposed in this paper and the general SWIPT scheme. The frame structure is also described in this section. Section 2.3 introduces the SNR constraint of a received signal, which is not considered in the related works.

### 2.1. Network Model

In general, a WPCN includes a HAP and *n* wireless stations, denoted as $s_{i}$, i=1,⋯,n, each equipped with a single antenna, as shown in Figure 1. Without loss of generality, the HAP is denoted as $s_{0}$. Since the HAP may be a mobile device, such as a mobile phone or a smart bracelet, therefore, it is assumed that the HAP is energy limited. The network follows the HTT protocol [8] and adopts the TDMA protocol for access control. By the HTT protocol, the time is divided into periods. The duration of a period is denoted as *T*. A period contains an ET phase and an IT phase. All wireless stations harvest energy during the ET phase and use the obtained energy for information transmission in the IT phase. The duration in the ET phase is denoted as $\tau_{0}$, and the IT phase is divided into *n* slots, each for a wireless station. The duration of each time slot is denoted as $\tau_{i}$. On the other hand, the transmission order of wireless stations is still an open issue and is out of the scope of the paper. The paper assumes the transmission order of wireless stations is according to the distance of the wireless station from the HAP. The closer, the earlier. Without loss of generality, the transmission order of wireless stations is assumed to be $s_{1},\ldots ,s_{n}$. In addition, we assume that the HAP and each wireless station have the Channel State Information (CSI) [5]. Due to the path loss and multipath effects, a wireless channel between the sender *s* and the receiver *r* with distance $d_{s,r}$ has a complicated channel model presented by CSI. For clear presentation, a channel gain coefficient *g* in Equations (Equation 2) and (Equation 6) is defined as the production of some constant parameters of CSI and the reciprocal of $d_{s,r}$ to an exponent λ. Hence, the received power can be obtained from *g* times *P*, where *P* is the power of the sender. In the ET phase, the channel gain coefficient between the HAP ($s_{0}$) and $s_{i}$ is denoted as $h_{i}$. In the IT phase, the channel gain coefficient between $s_{i}$ and the HAP ($s_{0}$) is denoted as $g_{i,0}$, and the channel gain coefficient between $s_{i}$ and $s_{j}$ is denoted as $g_{i,j}$. We also assume that the channels in the ET and IT phases are quasi-static fading channels [20]. That is, the channel coefficients $h_{i}$ and $g_{i,j}$ will not be changed during a period and can be regarded as a constant [16,17].

The wireless stations in the network can only obtain energy through the energy harvesting technology and will not have an additional internal energy supplement. In addition to the energy used for transmission during each period, wireless stations will also reserve a certain amount of energy to maintain basic operations, such as receiving beacons and modulating information. In addition, each wireless station, say $s_{i}$, has its throughput requirement, say $R_{i}^{th}$.

### 2.2. SWIPT vs. DT-SWIPT

With the advancement of antennas, a technology called SWIPT (Simultaneous Wireless Information and Power Transfer) [21,22] has emerged, which allows the sender to simultaneously transmit the energy signals and the information signals to the receiver. For example, in the ET phase, the HAP can transmit both the energy and information signals to the wireless stations at the same time, and the wireless stations can harvest energy and decode information. The SWIPT scheme can be classified mainly into two types: Time Switching (TS) and Power Splitting (PS), which are shown in Figure 2a and Figure 2b, respectively. When the antenna receives the signal, TS can switch the circuit to direct the signal to the charging circuit or the information receiving circuit by the time. On the other hand, when the antenna receives signals, in the PS type, the signals are directed to the power splitter, and the received signals are directed to the charging circuit and the information receiving circuit according to different proportions through the power splitter.

In the SWIPT scheme, the transmitter transmits the energy and information signals to the same receiver. This paper proposes another scheme called Different Target SWIPT (DT-SWIPT). In the DT-SWIPT scheme, the information and the energy transmitted by $s_{i}$ (i≠0) can be received by different receivers. That is, the information can be transmitted to the HAP, and the energy can be transferred to $s_{j}$ (j≠0 and j≠i). Since $s_{j}$ does not need to receive data from $s_{i}$, we adopt the TS type in our DT-SWIPT scheme. For example, suppose that there are an HAP and two wireless stations, say $s_{1}$ and $s_{2}$ in the network, as shown in Figure 3a. $s_{1}$ first transmits information to the HAP. When $s_{1}$ completes the information transmission, it can share the excess energy to $s_{2}$, as shown in Figure 3b.

Figure 4 is the timing diagram of the transmission of $s_{1}$ in the above example. Assume that $\tau_{1}$ is the transmission time of $s_{1}$. In $\tau_{1}$, there will be a ratio of $\alpha_{1}$ for information transmission, and the remaining $1-\alpha_{1}$ is used to share excess energy to other stations which have not transmitted to the HAP.

According to the above examples, the frame structure proposed in this paper is different from that of the general HTT protocol. The difference is that the wireless station will share its energy during $\left(1-\alpha_{i}\right)\tau_{i}$. Figure 5 is the frame structure proposed in this paper. Before a period starts, the HAP will first allocate its own transmission time ($\tau_{0}$), the transmission time of $s_{i}$ ($\tau_{i}$) and the proportion of time that $s_{i}$ used to transmit information ($\alpha_{i}$). Then, the HAP will send a beacon containing these parameters to all wireless stations. At the beginning of a period, the HAP will first transfer the energy to all wireless stations. At the end of the ET phase, all wireless stations will transmit information to the HAP sequentially. When the time of $\alpha_{i}*\tau_{i}$ has passed, $s_{i}$ will switch the signal circuit and start sharing the energy. The wireless station must consume all of its energy for transmission at the end of the period. Therefore, the wireless station in an earlier transmission order will not get the additional energy shared by other wireless stations that transmit later. As a result, the last transmitted station, $s_{n}$, will not share the energy. That is, $\alpha_{n}=1$.

### 2.3. SNR Constraint

In a general communication network, if the SNR value of a received signal is too low, it may affect the transmission speed and even cause decoding problems. Therefore, a successful transmission requires that the SNR value of the received signal should be larger than or equal to a certain SNR threshold. If the SNR value of the received signal is smaller than the SNR threshold, the signal may be regarded as a noise [23,24]. In a general data network, the SNR value of a received signal needs to be larger than 20 dB [18].

In summary, when a station receives a signal, the SNR value of the received signal must be larger than or equal to a certain threshold so that the signal can be decoded correctly. Nevertheless, none of the related works considers this issue. Therefore, it is impractical to apply the related works in real situations. Thus, in this paper, we take the SNR constraint of the HAP into account.

## 3. DT-SWIPT Assisted Time Allocation for the ECm Problem

The goal of this paper is to minimize the network energy consumption with the aid of the DT-SWIPT scheme. This problem is termed D-ECm in the paper. In this section, we will construct a non-linear programming model for the D-ECm and use an SQP algorithm to solve this optimization problem.

### 3.1. Problem Formulation

According to the HTT protocol, at the beginning of a period, the HAP will transfer energy to all wireless stations. The power used by the HAP is denoted as $P_{0}$. There is an upper limit, $P_{max}$, for the transmission power. The constraint of $P_{0}$ is shown in Equation (Equation 1).
(1)$$0\leq P_{0}\leq P_{max}.$$

In the ET phase, the channel gain between the HAP ($s_{0}$) and $s_{i}$ is denoted as $h_{i}$, which can be expressed in Equation (Equation 2).
(2)$$h_{i}=c_{0}d_{0,i}^{-\lambda },$$
where $d_{0,i}$ is the distance between the HAP ($s_{0}$) and $s_{i}$. Other parameters, such as antenna gain, antenna height of the HAP, etc., are neglected in the paper and termed $c_{0}$. For simplicity, $c_{0}$ is assumed to be a constant in the paper [23]. λ is the path loss exponent [25].

With the channel gain in Equation (Equation 2), when the HAP transfers energy in the ET phase, the received signal power of $s_{i}$ is denoted as $P_{0,i}^{r}$, which can be expressed in Equation (Equation 3).
(3)$$P_{0,i}^{r}=P_{0}h_{i}.$$

By Equation (Equation 3), the energy that $s_{i}$ can obtain in the ET phase is denoted as $E_{i}$, which can be expressed in Equation (Equation 4).
(4)$$E_{i}=\eta_{i}P_{0,i}^{r}\tau_{0},$$
where $\eta_{i}$ is the energy conversion efficiency of $s_{i}$ when receiving energy during energy harvesting, and $0\leq \eta_{i}\leq 1$. $\tau_{0}$ is the duration of the ET phase.

In addition to the energy required to transmit information to the HAP, wireless stations also need the energy for basic operations, such as receiving beacons, modulating information, and maintaining basic circuits. Therefore, among the obtained energy $E_{i}$, the energy of $s_{i}$ will have a proportion of $\rho_{i}$ for transmission, and the remaining $1-\rho_{i}$ energy will be used for basic operations, where $0\leq \rho_{i}\leq 1$. Thus, through Equation (Equation 4), the transmission power of $s_{i}$, denoted as $P_{i}$, can be expressed as Equation (Equation 5).
(5)$$P_{i}=\frac{\rho_{i}E_{i}}{\tau_{i}}.$$

According to the DT-SWIPT scheme, $s_{i}$ completes the transmission of information during $\alpha_{i}\tau_{i}$ and then shares the energy with other wireless stations during the remaining time, that is, $\left(1-\alpha_{i}\right)\tau_{i}$. Let $g_{i,j}$ be the channel gain coefficient from $s_{i}$ to $s_{j}$ (assuming that $s_{i}$ is transmitted earlier than $s_{j}$), which can be expressed in Equation (Equation 6).
(6)$$g_{i,j}=c_{i}d_{i,j}^{-\lambda },$$
where $d_{i,j}$ is the distance between $s_{i}$ and $s_{j}$. $c_{i}$ is a constant and stands for the other parameters, such as the antenna gain, antenna height of $s_{i}$, like $c_{0}$ in Equation (Equation 2).

Therefore, with the channel gain coefficient $g_{i,j}$, the additional energy shared from $s_{i}$ and obtained by $s_{j}$ is denoted as $E_{i,j}^{add}$, which can be expressed as Equation (Equation 7).
(7)$$E_{i,j}^{add}=\eta_{j}P_{i}g_{i,j}\left(1-\alpha_{i}\right)\tau_{i}.$$

By the TDMA protocol, wireless stations transmit information to the HAP sequentially. Therefore, each wireless station receives a different amount of energy shared by previously transmitted wireless stations. For this situation, let rank(i) be the transmission order of $s_{i}$, and $rank^{-1}\left(k\right)$ be the index of the wireless station whose transmission order is *k*. That is, if $s_{i}$ is the *k*th transmitting wireless station, then rank(i)=k and $rank^{-1}\left(k\right)=i$. If $s_{i}$ is the *m*th transmitting wireless station (1<k<m≤n), then $s_{i}$ will obtain the energy shared by $s_{rank^{-1}\left(1\right)},\ldots ,s_{rank^{-1}\left(k\right)},\ldots ,s_{rank^{-1}\left(m-1\right)}$. Therefore, the total additional energy obtained by $s_{i}$, which is shared from the previously transmitted wireless stations, is denoted as $E_{i}^{add}$, which can be expressed in Equation (Equation 8).
(8)$$E_{i}^{add}=\sum_{j=1}^{rank(i)-1}E_{rank^{-1}\left(j\right),i}^{add}.$$

In addition to the energy obtained from the HAP, $s_{i}$ also obtains additional energy shared by previously transmitted wireless stations. Therefore, the total energy obtained by $s_{i}$ is denoted as $E_{i}^{total}$, which can be expressed in Equation (Equation 9).
(9)$$E_{i}^{total}=E_{i}+E_{i}^{add}.$$

Since the first transmitting wireless station does not obtain additional energy, the wireless station with rank(i)=1 does not need to calculate the additional energy. As a result, the transmission power of $s_{i}$ will be replaced from Equation (Equation 5) to Equation (Equation 10).
(10)$$P_{i}=\{\begin{array}{cc} \frac{\rho_{i}E_{i}^{total}}{\tau_{i}}, & rank(i)=2,\ldots ,n,\\ \frac{\rho_{i}E_{i}}{\tau_{i}}, & rank(i)=1. \end{array}$$

Let $g_{i,0}$ be the channel gain coefficient from $s_{i}$ to the HAP ($s_{0}$), which can be expressed in Equation (Equation 11).
(11)$$g_{i,0}=c_{i}d_{i,0}^{-\lambda }.$$

While $s_{i}$ starts to transmit information to the HAP, with the channel gain coefficient $g_{i,0}$, the received signal strength of the HAP from $s_{i}$ is denoted as $P_{i,0}^{r}$, which can be expressed in Equation (Equation 12).
(12)$$P_{i,0}^{r}=P_{i}g_{i,0}.$$

To ensure that the HAP can decode the signal transmitted from $s_{i}$, the ratio of the received signal power $P_{i,0}^{r}$ to the noise power *N* must be larger than or equal to a certain threshold, which is denoted as $SNR_{HAP}^{th}$. The relationship is shown in Equation (Equation 13).
(13)$$\frac{P_{i,0}^{r}}{N}\geq SNR_{HAP}^{th}.$$

According to the Shannon’s Capacity Theorem [26], the channel capacity between HAP and $s_{i}$, which is denoted as $C_{i}$, can be expressed in Equation (Equation 14).
(14)$$C_{i}=B\log_{2}\left(1+\frac{P_{i,0}^{r}}{N}\right).$$

Obviously, $C_{i}$ will be affected by $P_{i,0}^{r}$ and the bandwidth *B*. According to Equation (Equation 14), the throughput of $s_{i}$ during the period *T* is denoted as $R_{i}$, which can be expressed in Equation (Equation 15).
(15)$$R_{i}=\frac{\alpha_{i}\tau_{i}}{T}C_{i}.$$

Since the work performed by each wireless station may be different, $s_{i}$ has its own throughput demand, $R_{i}^{th}$. The throughput of $s_{i}$ must be larger than or equal to $R_{i}^{th}$. The relationship between the $R_{i}^{th}$ and $R_{i}$ is shown in Equation (Equation 16).
(16)$$R_{i}\geq R_{i}^{th}.$$

This paper aims to minimize the energy consumption of the network under the conditions of Equations (Equation 13) and (Equation 16), for *i* from 1 to *n*. Since the energy source of wireless stations is harvested from the HAP, which can be considered as the energy consumption of the network. The energy consumption of the network is denoted as $E_{c}$, which can be expressed in Equation (Equation 17).
(17)$$E_{c}=P_{0}\tau_{0}.$$

From [20], it shows that when $P_{0}=P_{max}$, the energy consumption of the network will be the lowest. Therefore, let $P_{0}$ be $P_{max}$. By Equation (Equation 17), the goal will be to minimize the WET time, that is, $\tau_{0}$.

Based on the goal and constraints, a non-linear programming model for the D-ECm problem can be formulated in Model 1.

**Optimization Model 1** (D-ECm Problem)




$\underset{\begin{array}{c} \tau_{0},\tau_{1},\ldots ,\tau_{n}\\ \alpha_{1},\ldots ,\alpha_{n-1} \end{array}}{\mathrm{minimize}}$


*τ*
_0_
subject toC_1_ :Equation (Equation 13),    *i* = 1,…,*n*C_2_ :Equation (Equation 16),    *i* = 1,…,*n*C_3_ :0 ≤ *τ*_0_, *τ*_*i*_ ≤ *T*,    *i* = 1,…,*n*C_4_ :*τ*_0_ + $\sum_{i=1}^{n}$
*τ*_*i*_ ≤ *T*C_5_ :0 ≤ α_*i*_ ≤ 1,    *i* = 1,…,*n* − 1


In Model 1, $C_{1}$ denotes the *SNR constraint*. That is, the ratio of the received signal power $P_{i,0}^{r}$ to the noise power *N* must be greater than $SNR_{HAP}^{th}$ to ensure that the HAP can decode the signal transmitted from $s_{i}$. $C_{2}$ denotes the *throughput constraint*, which requires that the throughput of $s_{i}$ must meet its throughput demand $R_{i}^{th}$. $C_{3}$ denotes that the transmission time of the HAP and $s_{i}$ can not be longer than *T* and should be larger than 0. $C_{4}$ denotes that the sum of the transmission time of the HAP and wireless stations can not be longer than *T*. $C_{5}$ denotes that the DT-SWIPT ratio of $s_{i}$ during the transmission phase should be smaller than 1.

Since Model 1 is a nonlinear optimization problem, we use an SQP (Sequential Quadratic Programming) method [27] to obtain the optimal solution.

### 3.2. SQP-Based Algorithm for Finding the Optimal Solution of the D-ECm Problem

SQP [27] is an iterative method for solving constrained nonlinear optimization problems. SQP obtains the optimal solution by solving a sequence of QP subproblems. The standard form of the QP problem is formulated in Model 2. In Model 2, $\nabla f(\bar{x})$ and ${\left[\nabla f\left(\bar{x}\right)\right]}^{Tr}$ represent the gradients of the function $f(\bar{x})$ and the transpose matrix of $\nabla f(\bar{x})$, respectively. *M* and *N* represent the numbers of constraints of the equations and inequalities, respectively. The QP problem can get the corresponding optimal solution *w* according to the input $\bar{x}$. We can then take $\bar{x^{\prime }}=\bar{x}+w$ into the original QP problem to form a new QP problem and obtain another optimal solution $w^{\prime }$. Therefore, *w* can be regarded as the iterative step in SQP. When $\bar{x^{\prime }}-\bar{x}$ approaches 0, the corresponding $\bar{x}$ is the optimal solution of the SQP problem.

**Optimization Model 2** (Standard Form of the QP Problem)




$\underset{w}{\mathrm{minimize}}\;{\left[\nabla f(\bar{x})\right]}^{\mathrm{Tr}}w+\frac{1}{2}w^{T}\nabla^{2}f(\bar{x})w$

subject to

$h_{m}(\bar{x})+{\left[\nabla h_{m}(\bar{x})\right]}^{\mathrm{Tr}}w=0,\;\;\;m=1,\ldots ,M,$



$g_{n}(\bar{x})+{\left[\nabla g_{n}(\bar{x})\right]}^{\mathrm{Tr}}w\leq 0\;\;\;n=1,\ldots ,N,$




To satisfy the standard form of the QP problem, we turn the objective function of Model 1 into minimize $\tau_{0}^{2}$, and let $X=[\tau_{0},\tau_{1},\ldots ,\tau_{n},\alpha_{1},\ldots ,\alpha_{n-1}]$, where $X\in R^{2n+}$ and $R^{+}$ represent the set of non-negative real numbers. As a result, we rewrite Model 1 as Model 3.

**Optimization Model 3** (Reduced D-ECm Problem)



$\underset{X}{\mathrm{minimize}}\;F(X)=\tau_{0}^{2}$

subject toC_6_ :$R_{i}(X)=R_{i}^{\mathrm{th}}-R_{i}\leq 0,$*i* = 1,…, *n*,C_7_ :$S_{i}(X)={\mathrm{SNR}}_{\mathrm{HAP}}^{\mathrm{th}}-\frac{P_{i,0}^{r}}{N}\leq 0$*i* = 1,…, *n*,C_8_ :$T_{0}(X)=\tau_{0}-T\leq 0,$
C_9_ :$T_{i}(X)=\tau_{i}-T\leq 0,$*i* = 1,…, *n*,C_10_ :$T^{\prime }(X)=\tau_{0}+\sum_{i=1}^{n}\tau_{i}-T\leq 0,$
C_11_ :$A_{i}(x)=\alpha_{i}-1\leq 0,$*i* = 1,…, *n*, *n* − 1

Model 3 can be converted into a QP problem, called QP-ECm, shown in Model 4. In Model 4, $w^{k}$ is the optimal solution corresponding to $X^{k}$, $w^{k}\in R^{2n+}$. *k* is an iteration counter of the SQP.

**Optimization Model 4** (QP-ECm Problem)



$\underset{w^{k}}{\mathrm{minimize}}\;{\left[\nabla f(X^{k})\right]}^{T}w^{k}+\frac{1}{2}{\left(w^{k}\right)}^{T}\nabla^{2}f(X^{k})w^{k}$

subject to

C_12_:${\left[\nabla R_{i}(X^{k})\right]}^{\mathrm{Tr}}w^{k}\leq -R_{i}(X^{k}),$*i* = 1,…,*n*,C_13_:${\left[\nabla S_{i}(X^{k})\right]}^{\mathrm{Tr}}w^{k}\leq -S_{i}(X^{k}),$*i* = 1,…,*n*,C_14_:${\left[\nabla T_{0}(X^{k})\right]}^{\mathrm{Tr}}w^{k}\leq -T_{0}(X^{k}),$C_15_:${\left[\nabla T_{i}(X^{k})\right]}^{\mathrm{Tr}}w^{k}\leq -T_{i}(X^{k}),$*i* = 1,…,*n*,C_16_:${\left[\nabla T^{\prime }(X^{k})\right]}^{\mathrm{Tr}}w^{k}\leq -T^{\prime }(X^{k}),$C_17_:${\left[\nabla A_{i}(X^{k})\right]}^{\mathrm{Tr}}w^{k}\leq -A_{i}(X^{k}),$*i* = 1,…,*n* − 1

To solve the D-ECm problem, an algorithm based on SQP is proposed, called DaTA (**D**T-SWIPT **a**ssisted **T**ime **A**llocation), which is shown in Algorithm 1. In line 2, we convert the ECm problem into a QP-ECm problem. In line 3, set the iterative counter *k* and select an initial point $X^{0}$ that meets all constraints. In line 5, solve the QP-ECm problem according to $X^{k}$ and get the optimal solution $w^{k}$. In lines 6 and 7, increase the iterative counter *k*, and let $X^{k}=X^{k-1}+w^{k-1}$. If $X^{k}-X^{k-1}$ approaches 0, return the optimal solution $X^{k-1}$; otherwise, return to line 5 to solve a new QP problem again.
**Algorithm 1:** DaTA: a SQP-based Algorithm for the D-ECm Problem
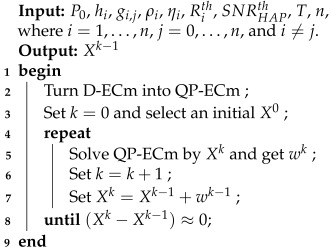


In this section, we have formulated a nonlinear optimization problem, called D-ECm, and found the optimal solution through a SQP-based algorithm, named DaTA. However, it costs much time to find the optimal solution. However, to our best knowledge, there is no heuristic method proposed in the literature to find a plausible solution. Therefore, we propose a heuristic method, called DaTA-H, and describe it in the following section.

## 4. DaTA-H: The Heuristic Method

In this section, we introduce the concept and the algorithm of the proposed heuristic method, named DaTA-H. In Section 3, we use SQP to solve the non-linear optimization problem to obtain the optimal solution. However, the number of wireless stations, *n*, will affect the number of variables to be determined. Moreover, a large *n* results in too many SQP iterations and causes a long calculation time. Therefore, we simplify the D-ECm problem by reducing the number of variables and make the result close to the optimal solution obtained by the SQP.

### 4.1. Problem Formulation of the Simplifying D-ECm Problem

In Section 3.1, we can know that the transmission time of $s_{i}$, $\tau_{i}$, is related to $\tau_{0}$ through Equations (Equation 4) and  (Equation 10). To simplify the D-ECm problem, we only take $\tau_{0}$ into account. The remaining time, $T-\tau_{0}$, is evenly distributed to all wireless stations. Therefore, in DaTA-H, the transmission time of $s_{i}$, i.e., $\tau_{i}$, can be expressed as Equation (Equation 18).
(18)$$\tau_{i}=\frac{T-\tau_{0}}{n},i=1,\ldots ,n.$$

By Equation (Equation 18), we can simplify variables such as $\tau_{1},\ldots ,\tau_{n}$ into a function that contains $\tau_{0}$.

Each $s_{i}$ uses a proportion $\alpha_{i}$ of $\tau_{i}$ to transmit information and the rest $1-\alpha_{i}$ to charge another station. To simplify the D-ECm problem and ease to the calculation, we further simplify $\alpha_{1},\ldots ,\alpha_{n-1}$ by means of changing the transmission order. According to Equations (Equation 2)–(Equation 4), the distance $d_{0,i}$ between the HAP ($s_{0}$) and $s_{i}$ will directly affect the energy obtained by $s_{i}$. The wireless station closest to the HAP can obtain the most energy, while the wireless station farthest from the HAP harvests the least. Therefore, we rearrange the order of the wireless stations according to $d_{0,i}$. The wireless station closest to the HAP is marked as $s_{1}$, the next is $s_{2}$, and so on. The farthest one is $s_{n}$. In DaTA-H, we let the wireless station that obtains the most energy from the HAP (i.e., $s_{1}$) be the first to transmit information and share the energy with other stations for the longest time. The wireless station that obtains the second most energy (that is, $s_{2}$) is the second to transmit information and share energy for the second longest time, and so on. The farthest wireless station (i.e., $s_{n}$) is the last one to transmit information and needs not to share energy. Hence, $\alpha_{n}=1$. In summary, the time ratio $\alpha_{i}$ of the DT-SWIPT can be allocated by an arithmetic sequence, which can be expressed as Equation (Equation 19).
(19)$$\alpha_{i}=\alpha +i*\frac{\alpha_{n}-\alpha }{n},i=1,\ldots ,n,$$
where α is a base value of the DT-SWIPT. Since α behaves differently due to the different SNR thresholds of the HAP, we assume that α is known in advance. Section 5 will discuss the impact of α on the performance.

According to the above summary, we can rewrite the D-ECm problem as an S-ECm (Simplified D-ECm) problem and is shown in Model 5.

**Optimization Model 5** (S-ECm: Simplified D-ECm Problem)

minimizeτ_0_subject toC_18_:$\frac{P_{i,0}^{r}}{N}\leq {\mathrm{SNR}}_{\mathrm{HAP}}^{\mathrm{th}},$*i* = 1,…,*n*C_19_:$R_{i}\leq R_{i}^{\mathrm{th}},$*i* = 1,…,*n*C_20_:$0\leq \tau_{0}\leq T$


Model 5 is still a non-linear programming problem due to $C_{19}$. Since the variable to be optimized is only $\tau_{0}$, we can use a linear search method to iteratively find an approximate solution of $\tau_{0}$.

### 4.2. DaTA-H Algorithm for S-ECm

The DaTA-H algorithm is shown in Algorithm 2. In addition to the inputs used in Algorithm 1, the inputs also include the α and the step size δ. Algorithm 2 is divided into two steps: the initialization step and the iterative step.

In the *Initialization Step*, because we need to find an approximate solution of $\tau_{0}$, we first set a variable *k* to count the number of iterations (line 2) and set the WET time $\tau_{0}^{k}$ in line 3.

**Lemma** **1.**
*Let $\tau_{0}^{th}$ be the minimum of the WET time and $\theta =\frac{SNR_{HAP}^{th}*N}{g_{1,0}\rho_{1}\eta_{1}h_{1}P_{0}+\frac{SNR_{HAP}^{th}*N}{n}}$. Then $\tau_{0}^{th}$ can be obtained as follows.*

$$\tau_{0}^{th}=\frac{T}{n}*\theta .$$



**Proof.** To simplify the D-ECm problem, we have proposed a simplified model, S-ECm. There we have assumed that the wireless station closest to the HAP is $s_{1}$, followed by $s_{2}$, and so on. Therefore, the minimum WET time should at least satisfy the energy requirement of $s_{1}$. As a result, $\tau_{0}^{th}$ can be regarded as the WET time required for $s_{1}$. By $C_{18}$, we can obtain Equation (Equation 20).
(20)$$P_{1,0}^{r}\geq SNR_{HAP}^{th}*N.$$By Equation (Equation 12), Equation (Equation 20) can be rewritten as below.
(21)$$P_{1}\geq SNR_{HAP}^{th}*\frac{N}{g_{1,0}}.$$Since $s_{1}$ is the first wireless station to transmit information to the HAP (i.e., rank(1)=1), the energy obtained by $s_{1}$ is totally from the HAP, no additional energy from any other wireless station. By Equation (Equation 10), Equation (Equation 21) is turned as follows.
(22)$$E_{1}\geq SNR_{HAP}^{th}*\frac{N\tau_{1}}{g_{1,0}\rho_{1}}.$$Furthermore, by Equations (Equation 3) and  (Equation 4), Equation (Equation 22) is turned as follows.
(23)$$\tau_{0}\geq SNR_{HAP}^{th}*\frac{N\tau_{1}}{g_{1,0}\rho_{1}\eta_{1}h_{1}P_{0}}.$$$\tau_{0}^{th}$ is the minimum value of $\tau_{0}$. By Equation (Equation 18), $\tau_{1}=\frac{T-\tau_{0}^{th}}{n}$. As a result, we have the following equation.
(24)$$\tau_{0}^{th}=\frac{SNR_{HAP}^{th}*N*T}{n*g_{1,0}\rho_{1}\eta_{1}h_{1}P_{0}+SNR_{HAP}^{th}*N}.$$Since $SNR_{HAP}^{th}$, *N*, *n*, $g_{1,0}$, $\rho_{1}$, $\eta_{1}$, $h_{1}$, and $P_{0}$ are all known parameters, let$\theta =\frac{SNR_{HAP}^{th}*N}{g_{1,0}\rho_{1}\eta_{1}h_{1}P_{0}+\frac{SNR_{HAP}^{th}*N}{n}}$. We simplify Equation (Equation 24) as below.
$$\tau_{0}^{th}=\frac{T}{n}*\theta ,$$
which concludes the proof of the lemma.    □

**Algorithm 2:** DaTA-H Algorithm for the S-ECm Problem

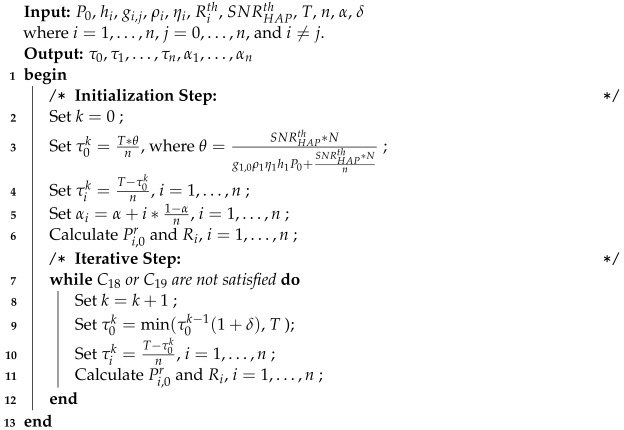



According to Lemma 1, we can know that if $s_{1}$ satisfies $C_{18}$, $\tau_{0}^{k}$ must be equal to $\tau_{0}^{th}$. To avoid $\tau_{0}^{k}$ starting from 0 and causing too many iterations, therefore, in the initialization step, we let $\tau_{0}^{k}$ iterate from $\tau_{0}^{th}$. According to Equation (Equation 18), line 4 is to evenly set $\tau_{i}^{k}$ for all wireless stations. Based on α, line 5 sets the DT-SWIPT time ratio $\alpha_{i}$. Therefore, we can derive $P_{i,0}^{r}$ and $R_{i}$, for i=1,…,n.

In the *Iterative Step*, we first check whether all wireless stations satisfy the constraints of $C_{18}$ and $C_{19}$ based on the derived $P_{i,0}^{r}$ and $R_{i}$ in the initialization step (line 7). If one of the constraints is not satisfied, the iterative step starts. Firstly, line 8 increases the iteration index *k*. Line 9 increases $\tau_{0}^{k}$ by adding a step size δ. According to Equation (Equation 18), line 10 sets a new transmission time for all wireless stations. Line 11 recalculates $P_{i,0}^{r}$ and $R_{i}$, i=1,…,n. Finally, go back to line 7 to recheck whether the constraints of $C_{18}$ and $C_{19}$ are satisfied for all wireless stations or not.

In the iterative step, there is an upper limit for the number of iterations. That is, the WET time $\tau_{0}^{k}$ must be smaller than or equal to *T* after several iterations. Therefore, we derive the upper bound of the number of iterations by Lemma 2.

**Lemma** **2.**
*In Algorithm 2, the number of iterations in the iterative step is not greater than $\log_{1+\delta }n-\log_{1+\delta }\theta$.*


**Proof.** Since the increase of the $\tau_{0}^{k}$ must be smaller than or equal to *T*, it bounds the number of iterations in the iterative step. In other words, we can have the following inequality.
$$\tau_{0}^{k}=\tau_{0}^{th}{\left(1+\delta \right)}^{k}\leq T.$$According to Lemma 1, $\frac{T}{n}\theta {\left(1+\delta \right)}^{k}\leq T$. As a result,
$$k\leq (\log_{1+\delta }n-\log_{1+\delta }\theta ),$$
which concludes the proof of the lemma. □

Furthermore, the time complexity of Algorithm 2 is analyzed in Theorem 1.

**Theorem** **1.**
*The time complexity of Algorithm 2 is O(nlogn).*


**Proof.** In the Initialization Step, the time to calculate $\tau_{0}^{k}$, $\tau_{i}^{k}$, $P_{i,0}^{r}$, and $R_{i}$ is O(1) and the numbers of calculations of $P_{i,0}^{r}$ and $R_{i}$ are respectively *n*. Therefore, the time complexity of the Initialization Step is O(n). In the Iterative Step, according to Lemma 2, the number of iterations of the while loop is $\log_{1+\delta }n-\log_{1+\delta }\theta$. The time to calculate lines 8 and 9 is O(1) and that to calculate lines 10 and 11 is *n*, respectively. Therefore, the time complexity of the Iterative Step is O(nlogn). As a result, the time complexity of Algorithm 2 is O(nlogn), which concludes the theorem. □

In this section, we have simplified the S-ECm problem and proposed a heuristic method, called DaTA-H. Based on the result of Lemma 1, we avoid $\tau_{0}^{k}$ iterating from 0 to cause too many iterations. We use a linear search method to find an approximate solution of $\tau_{0}$. In addition, we also prove that the time complexity of Algorithm 2 is O(nlogn)) in Theorem 1 based on Lemma 2. In the following section, we will compare the efficiency of DaTA and DaTA-H and show the advantages against the related work.

## 5. Performance Evaluations

In this section, we will assess the efficacy of the proposed techniques by comparing them with the corresponding studies. The SQP module in Matlab [27] is utilized to identify the best solutions for the associated studies and the techniques introduced in the paper. In addition, we evaluate the performance of these methods by considering factors such as energy consumption, sum throughput, and energy efficiency of the network.

The parameters used in the simulations are shown in Table 1. For ease of discussion, we assume that the energy harvest efficiencies, $\eta_{i}$, and the proportions of the energy for transmission, $\rho_{i}$, of each wireless station are the same in the simulations. Moreover, we also assume that $c_{0}$, $c_{i}$, and λ are the same for all wireless stations. The throughput demands $R_{i}^{th}$ for all wireless stations are the same. The period time, *T*, is normalized to 1 s.

In the simulations, a general scenario is designed to observe the efficiency of the proposed methods against the related works in terms of the energy consumption, the sum throughput, and the energy efficiency of the network. In addition, we further design a special scenario to observe the effect of the DT-SWIPT scheme. For both scenarios, the HAP is located at the center of the network. The coordinate of the HAP is (0, 0). The differences between the general and the special scenarios are the placements of the wireless stations. In the general scenario, 10 wireless stations are randomly deployed within 4 to 10 m from the HAP. However, in the special scenario, the 10 wireless stations are arranged in the same quadrant. In other words, suppose the coordinate of $s_{i}$ is ($x_{i}$, $y_{i}$), i=1,…,10. In the general scenario, the requirement for $x_{i}$ and $y_{i}$ is $4\leq \sqrt{x_{i}^{2}+y_{i}^{2}}\leq 10$, while, in the special scenario, it requires the coordinate of $s_{i}$ to be ($|x_{i}|$,$|y_{i}|$). In both scenarios, wireless stations transmit information to the HAP from near to far according to the distance to the HAP. The results are the average of 30 sets of data.

In the simulation, four different metrics are compared, including the network energy consumption, the total network throughput, the network energy efficiency, and the number of failed decoded signals. To observe the effects of the DT-SWIPT scheme, the methods proposed in the paper, DaTA and DaTA-H, are compared. In addition, the schemes proposed in the related work [8,10,16] are also compared, namely Sum-Throughput Maximization (STM) [8], Energy Efficiency Maximization (EEM) [10], and Energy Consumption minimization (ECm) [16].

### 5.1. Comparisons of the Network Energy Consumption

The network energy consumption is defined as the production of the power *P* of HAP and $\tau_{0}$ in the duration of the ET referred to Equation (Equation 17). Figure 6a and Figure 6b are the comparisons of the network energy consumption in general and special scenarios, respectively. In these two figures, it can be found that the energy consumption of the STM is the highest because the STM only considers how to maximize the throughput. As a result, wireless stations obtain higher energy to transmit and the signals transmitted to the HAP are strong. Therefore, the STM has a good performance in the metric of failed transmissions, which will be illustrated later. On the other hand, the energy consumption of the EEM and the ECm are almost the same respectively either in the general or in the special cases because both the two methods take the energy consumption into consideration. In addition, both the DaTA and the DaTA-H use the DT-SWIPT scheme to allow wireless stations sharing excess energy to others. Therefore, the DaTA performs better than the EEM and the ECm in terms of the energy consumption. Since the DaTA-H is a heuristic method and the transmission time of each wireless station is equal, in the general scenario, the energy consumption of the DaTA-H is not as good as those of the EEM and the ECm. However, in the special case, due to the wireless stations are close to each other, the DaTA-H that use the DT-SWIPT scheme has better results than the EEM and the ECm in the energy consumption when the SNR threshold is 25 dB. Moreover, in the DaTA-H, there is no change in energy consumption when the SNR threshold is between 20 and 25 dB in both the general and the special scenarios. The reason is that when the wireless station meets the throughput demand, its transmission power also meets the SNR threshold. As the SNR threshold increases, the power demanded by wireless stations increases and the network energy consumption also increases.

In Section 4, to simplify the D-ECm problem, we have introduced a variable α to ease the calculation of the time ratio for information transmission, $\alpha_{i}$, when using the DT-SWIPT scheme. To observe the impact of α on the performance of the DaTA-H scheme, Figure 7 is the comparisons of the network energy consumption of the DaTA-H scheme in the general (Figure 7a) and the special (Figure 7b) scenarios when α is 0.5, 0.7, and 0.8, respectively. When α is 0.5, the wireless station dedicates more time to energy sharing compared to when α is 0.7 or 0.8. Therefore, the proportion of transmission time that it takes to transmit data decreases. In addition, when α are 0.7 and 0.8, if the SNR threshold reaches a certain value, the energy consumption begins to increase. When α is 0.7, the time ratio of data transmission by the wireless station is less than that when α is 0.8. Therefore, the wireless station needs more energy to meet the throughput demand. With an increase in the SNR threshold, the energy consumption when α is 0.7 will be less than that when α is 0.8. The reason is that when α is 0.7, the wireless stations farther away from the HAP will receive more energy than that when α is 0.8. In addition, the signal power received from the wireless station to the HAP is easier to reach the SNR threshold, so it consumes more energy when α is 0.8.

### 5.2. Comparisons of the Network Sum-Throughput

The Network Sum-Throughput is the total throughput of all wireless stations during the HTT Period. Figure 8a and Figure 8b are comparisons of the network sum-throughput in the general and the special scenarios, respectively. In these two figures, STM has the highest sum-throughput because the purpose of this model is to maximize the sum-throughput of the network. In Figure 6a,b, the energy consumption of the ECm and the EEM is almost the same, but in Figure 8a,b, the sum-throughput of the ECm is the highest. The reason is that the EEM considers the energy efficiency of wireless stations. Therefore, to maximize the total energy efficiency of wireless stations, the throughput of some wireless stations will be less. Therefore, the sum-throughput of the EEM is lower than that of the ECm. In the DaTA scheme, the sum-throughput of the general scenario is better than that in the special scenario. The reason is that, in the general scenario, the deployment of wireless stations is relatively scattered. Therefore, the proportion of time that wireless stations share energy will be less than that in the special scenario due to the poor efficiency. In the special scenario, the sum-throughput of the DaTA scheme is quite close to the throughput demand. The wireless stations in DaTA-H have the same transmission time. As a result, the sum-throughput of the DaTA-H scheme is directly affected by the network energy consumption. In addition, Figure 8a,b also show that the sum-throughput of the DaTA scheme is irregular. The reason is that the goal of the DaTA scheme is to minimize the ET (energy transfer) time of the HAP. Therefore, there may be multiple sets of optimal solutions that can obtain the same ET time.

Figure 9a and Figure 9b illustrate the comparisons of the network sum-throughput of the DaTA-H scheme with different α values in the general and the special scenarios, respectively. When α is 0.8, the sum-throughput of α equal to 0.8 is higher than that when α is 0.5 and 0.7. It is because, in the case of α is 0.8, the time ratio of the wireless station to transmit data is longer than that when α is 0.5 and 0.7. Even though the wireless station can obtain more energy when α=0.5, its sum-throughput is still lower than that when α equals 0.7 or 0.8 due to a larger proportion of time for energy sharing. In addition, when the SNR threshold is within a certain range, the sum-throughput of the DaTA-H scheme is almost the same. The reason is that all the wireless stations share the energy with a fixed proportion of the transmission time. The wireless stations farther from the HAP can better meet the SNR threshold. Therefore, the wireless stations only need to meet the throughput demand. In Figure 9b, wireless stations are close to each other in the special case, wireless stations can obtain more extra energy from others. However, the DaTA scheme uses an optimized method to obtain the result of time allocation. Therefore, the IT time allocated for the wireless station far from the HAP is just enough for the station to transmit the data in demand. As a result, the sum-throughput of the DaTA scheme is not as good as that of the DaTA-H scheme.

### 5.3. Comparisons of the Network Energy Efficiency

Figure 10 and Figure 10b illustrate the comparisons of the network energy efficiency (EE) in the general and the special scenarios, respectively. Regarding the EE, we consider the ratio of the network sum-throughput to the network energy consumption, instead of considering the energy efficiency from the wireless station viewpoint [10]. The calculation of the energy efficiency is as follows.
(25)$$EE=\frac{Network\_Sum\_Throughout}{Network\_Energy\_Consumption}.$$

In Figure 10, the STM has the worst energy efficiency. Although the STM has the best network throughput, the energy consumption of the STM is quite high. Therefore, its energy efficiency is poor. Besides, the energy efficiency of the ECm is slightly better but close to that of the EEM. Although the EEM is aimed at energy efficiency [10], the EEM considers the energy efficiency of the wireless station itself rather than the entire network. Thus, according to Figure 6 and Figure 8, the energy efficiency of the ECm is slightly better than that of the EEM. On the other hand, the sum-throughput of the DaTA is worse than those of the ECm and the EEM. However, through the DT-SWIPT scheme, the energy consumption of the DaTA is less than those of the ECm and the EEM. Therefore, the DaTA is superior in terms of the energy efficiency. Since wireless stations are close to each other in the special scenario, even though the sum-throughput of the DaTA is not the highest, the energy efficiency of the DaTA is much better than those of the ECm and the EEM due to the benefits of the DT-SWIPT scheme. In the general scenario, the placement of wireless stations is scattered and the transmission time of the wireless stations is the same. Therefore, the energy consumption of the DaTA-H is the highest, which leads to a poor energy efficiency. However, in the special scenario, wireless stations can obtain more energy shared by others through the DT-SWIPT scheme. Therefore, as the SNR threshold increases, the energy efficiency of the DaTA-H is gradually better than those of the ECm and the EEM.

Figure 11a,b are the comparisons of the energy efficiency of the DaTA-H scheme with different α in the general and the special scenarios, respectively. The energy efficiency is the worst in the case of α equal to 0.5 because the energy consumption when α is 0.5 is much higher than those when α are 0.7 and 0.8. In the general scenario, the energy consumption and the sum-throughput in the case of α equal to 0.8 perform better than those when α=0.7. Therefore, the energy efficiency when α=0.8 is better than that when α=0.7. However, in the special scenario, since wireless stations can share much energy with others when α=0.7, even the sum-throughput is not as good as that when α=0.8, the energy efficiency when α=0.7 performs better than that when α=0.8 as the SNR threshold increases.

### 5.4. Comparisons of the Failed Decoded Signals

Figure 12 is the comparisons of the number of failed decoded signals. In Figure 12, the DaTA and the DaTA-H schemes both take the SNR threshold into account, but the STM, the EEM, and the ECm schemes do not. For the schemes considering the SNR threshold, the number of failed decoded signals of these schemes are 0. However, since the goal of the STM scheme is to maximize the network sum-throughput, therefore, the power used by the wireless station of the STM scheme is large enough to satisfy the SNR constraint. As a result, the STM scheme is less susceptible to the changes in the SNR threshold.

The goals of the EEM and the ECm schemes consider energy efficiency [10] and the minimization of the energy consumption, respectively. Therefore, in terms of time allocation, the ET time for the HAP to transmit energy will be short, and wireless stations will receive less energy. Therefore, the number of failed decoded signals of the EEM and the ECm schemes will gradually increase as the SNR threshold increases. This figure also verifies the effect of the model considering the SNR constraint.

### 5.5. Summary

Table 2 summarizes the above performance comparison results in general scenarios. The DaTA scheme has good performance in both the special and the general scenarios in terms of energy consumption and energy efficiency. On the other hand, although the heuristic method, DaTA-H, performs not good in the general scenario, the DaTA-H scheme performs well in the special scenario in terms of energy consumption and energy efficiency. By the evaluation of the DaTA scheme, the advantages of the DT-SWIPT mechanism are verified. In the special scenario, the superiority of the DT-SWIPT scheme is more evident. Furthermore, the energy consumption of the DaTA scheme compared to the ECm scheme is 20% less in the general scenario and is more than 60% less in the special scenario. The heuristic method, DaTA-H, has better results than the ECm and the EEM schemes in the special scenario when the SNR threshold is getting large. This phenomenon also reveals the benefit of applying the DT-SWIPT scheme to wireless powered communication networks. From the simulation results, it is critically important that the SNR threshold have to be taken into consideration when establishing the optimization model. Otherwise, even if the optimal result can be obtained, it will fall into the dilemma of not being able to decode the received signal in reality. Consequently, the paper is the first to consider this issue and the schemes proposed in the paper perform better than the related work.

## 6. Conclusions

This paper studied the DT-SWIPT assisted wireless powered communication network and solved the energy consumption minimization problem through time allocation. According to the HTT protocol, we proposed a new frame structure and, based on the frame structure, established an optimization model. To solve the non-linear programming problem, we used the SQP algorithm to obtain the optimal solution. In addition, this paper also proposed a heuristic method to obtain a feasible solution by simplifying the variables in the original problem. In performance evaluation, the DaTA scheme proposed in this paper can effectively save the network energy consumption in both the general and the special scenarios due to the DT-SWIPT scheme. If wireless stations are deployed closely, the DaTA-H scheme can perform quite well. This result shows that the DT-SWIPT scheme can improve energy efficiency effectively. Furthermore, in the optimization model, we take the SNR constraint into account. The performance results also verify the importance of the effect of the SNR constraint, which was not considered by the previous work. Since QoS and mobility are important issues, especially for critical applications, we will take the QoS and/or mobility issues into account in the near future.

## Figures and Tables

**Figure 1 sensors-24-05535-f001:**
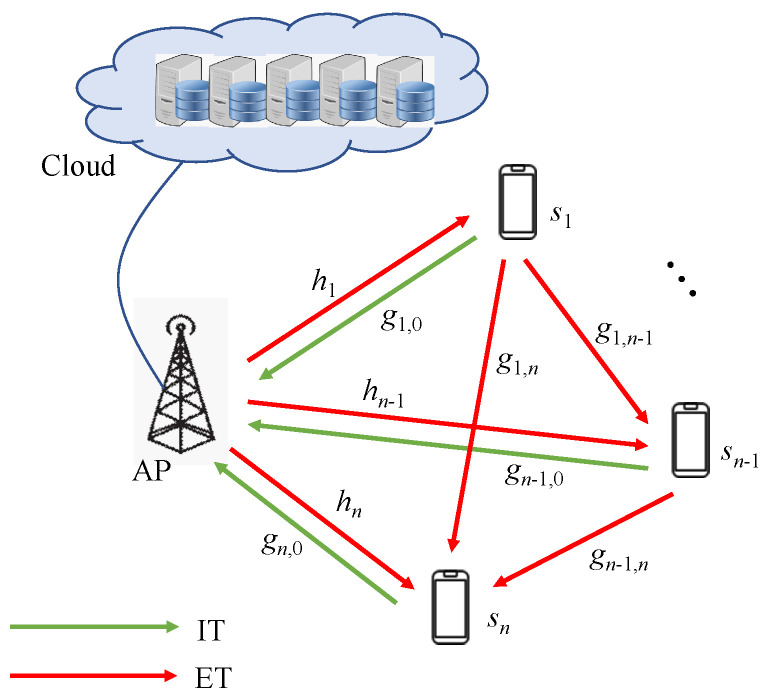
The network model.

**Figure 2 sensors-24-05535-f002:**
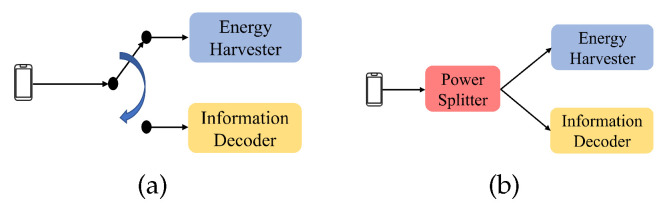
Different types of the SWIPT technology. (**a**) Time Switching. (**b**) Power Splitting.

**Figure 3 sensors-24-05535-f003:**
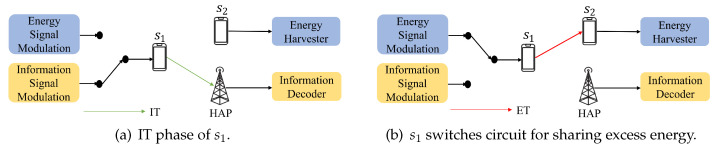
DT-SWIPT scheme switches between different receivers.

**Figure 4 sensors-24-05535-f004:**
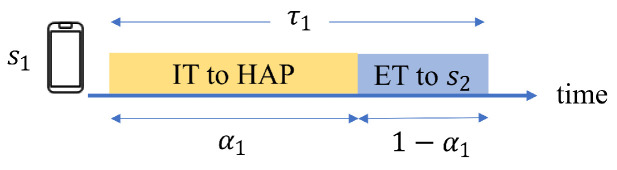
The timing diagram of $s_{1}$.

**Figure 5 sensors-24-05535-f005:**
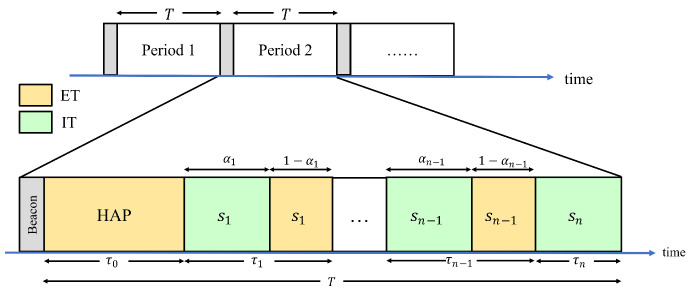
The frame structure.

**Figure 6 sensors-24-05535-f006:**
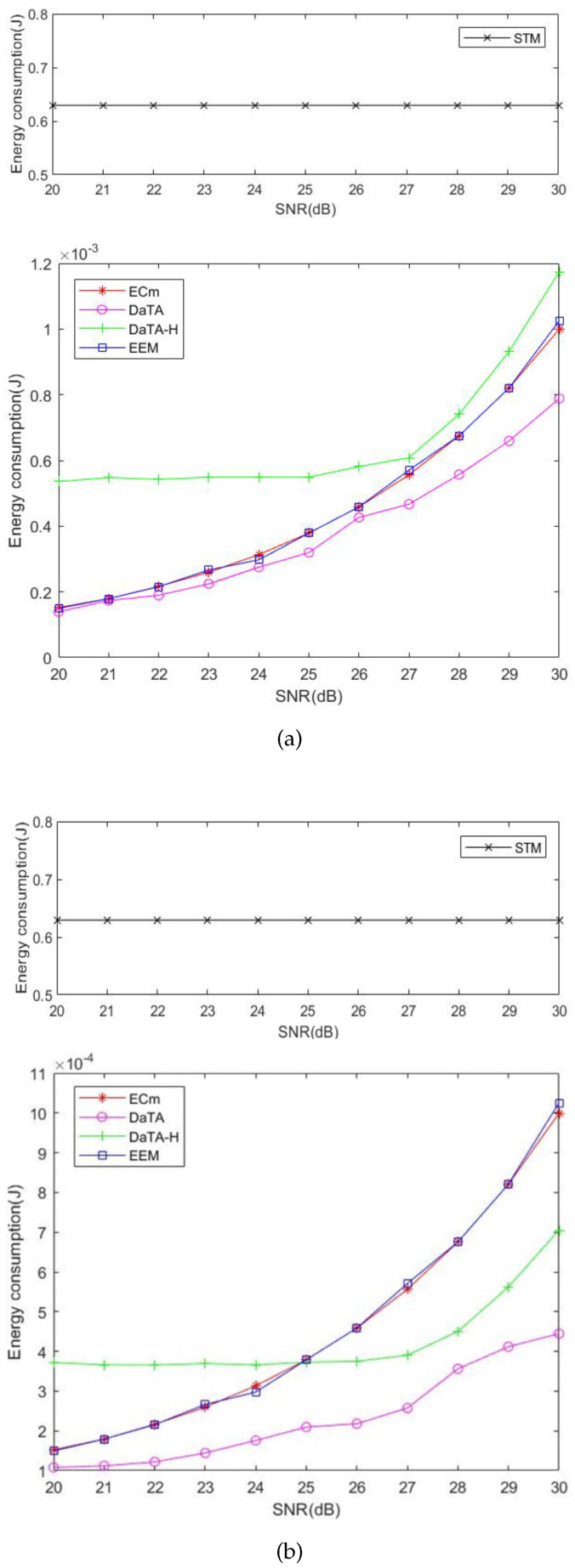
Comparisons of the network energy consumption. (**a**) The general scenario. (**b**) The special scenario.

**Figure 7 sensors-24-05535-f007:**
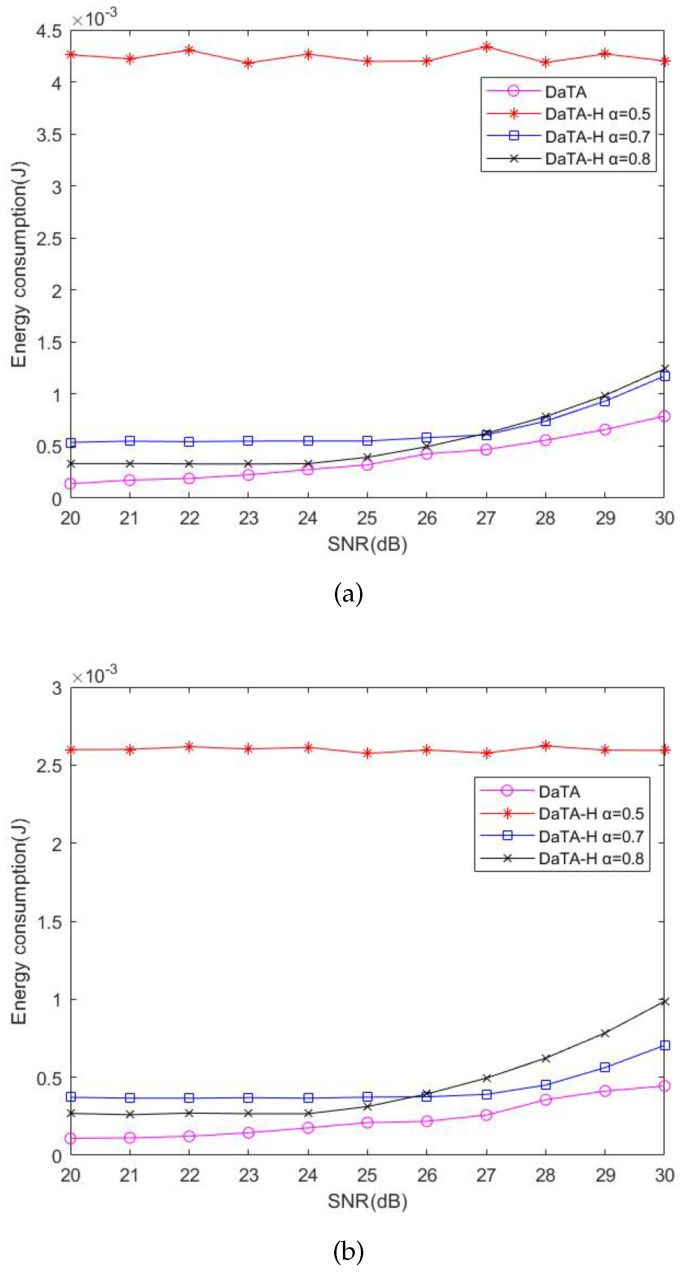
Comparisons of network energy consumption of the DaTA-H scheme with different α. (**a**) The general scenario. (**b**) The special scenario.

**Figure 8 sensors-24-05535-f008:**
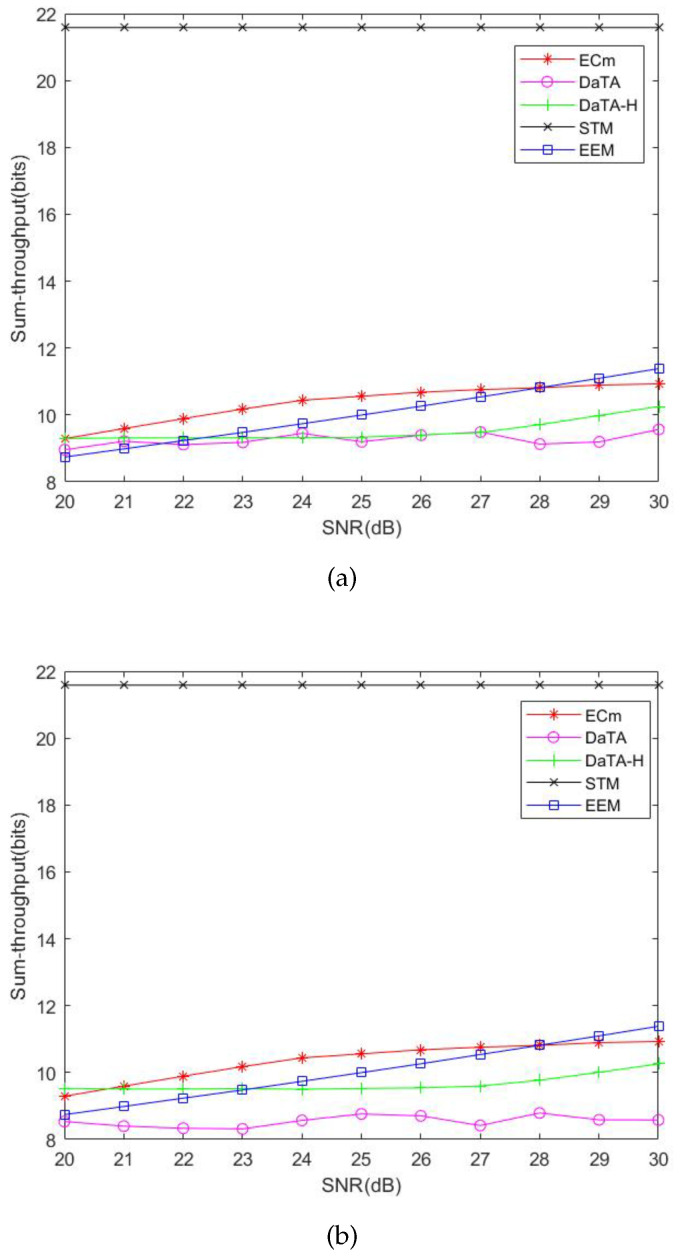
Comparisons of the network sum-throughput. (**a**) The general scenario. (**b**) The special scenario.

**Figure 9 sensors-24-05535-f009:**
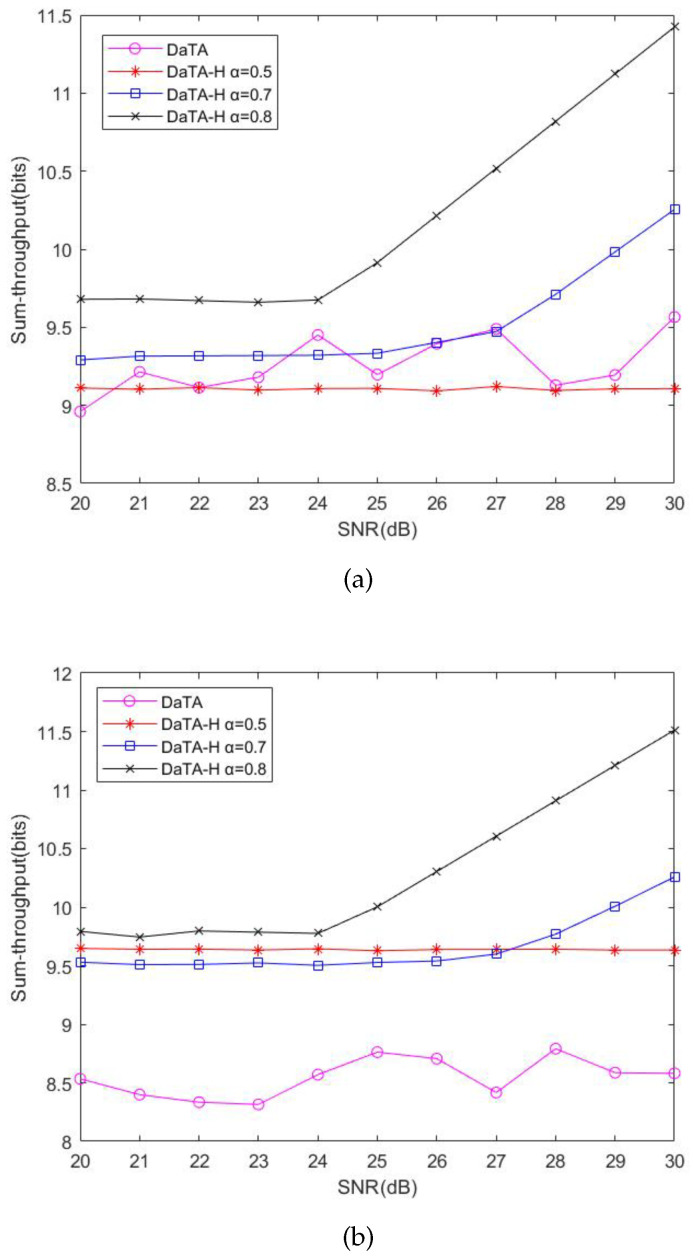
Comparisons of the network sum-throughput of the DaTA-H scheme with different α. (**a**) The general scenario. (**b**) The special scenario.

**Figure 10 sensors-24-05535-f010:**
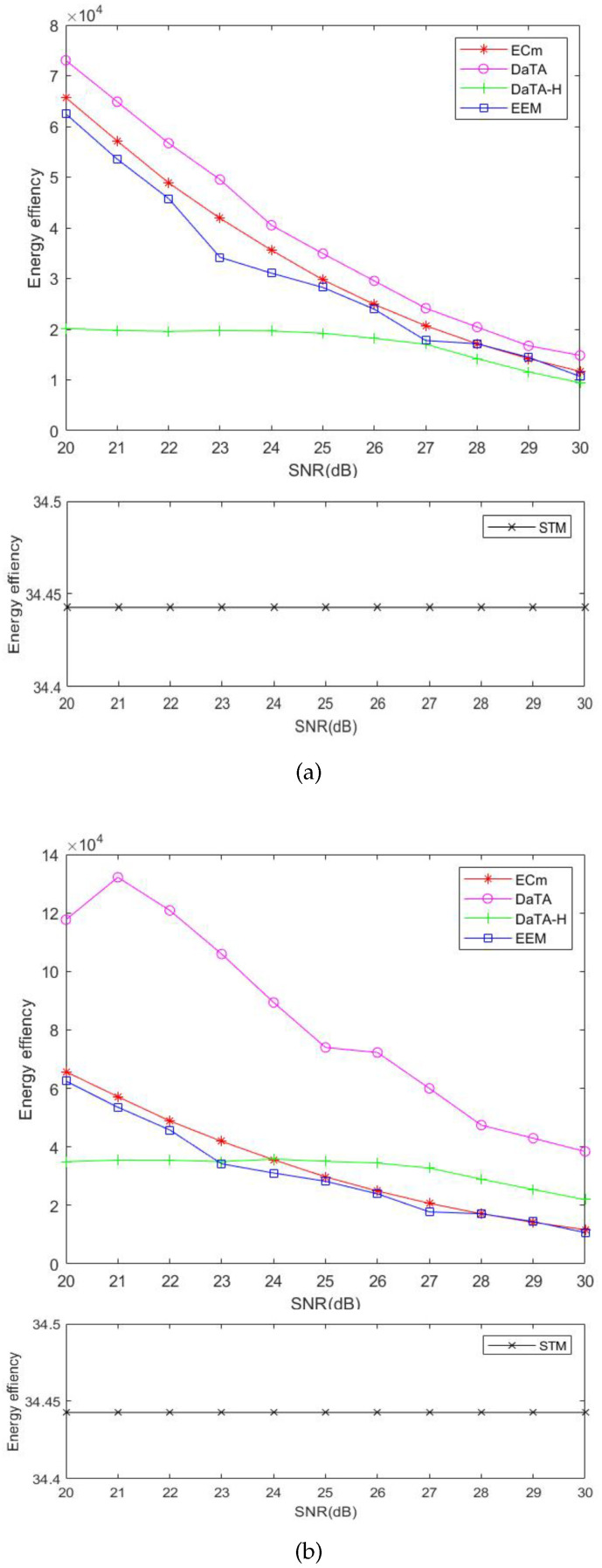
Comparisons of the network energy efficiency. (**a**) The general scenario. (**b**) The special scenario.

**Figure 11 sensors-24-05535-f011:**
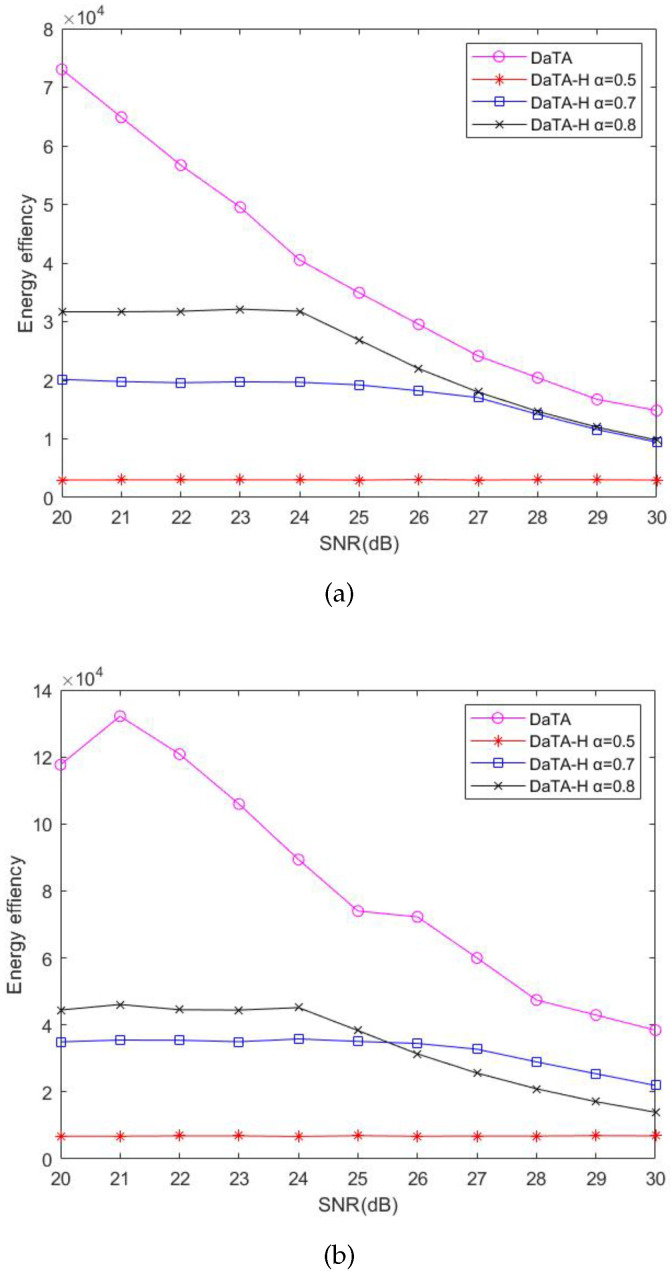
Comparisons of the network energy efficiency of the DaTA-H scheme with different α values. (**a**) The general scenario. (**b**) The special scenario.

**Figure 12 sensors-24-05535-f012:**
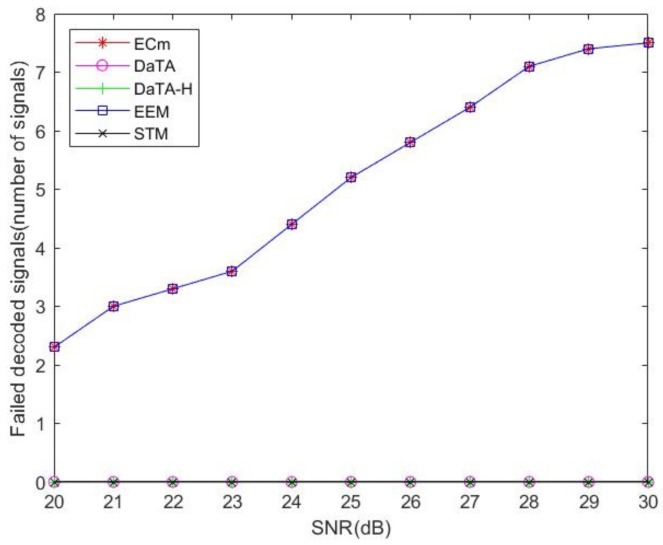
Comparisons of the number of failed transmissions.

**Table 1 sensors-24-05535-t001:** Simulation Settings.

Parameter	Value
$\eta_{i}$, energy harvest efficiency	1 [14]
$\rho_{i}$, the portion of energy for IT	1
$P_{0}$, the transmission power of the HAP	30 dBm [20]
*N*, noise power	−70 dBm [14]
λ, path loss exponent	3 [14]
$R_{i}^{th}$, throughput demand	0.8 bps
α (in DaTA-H)	0.7
δ (in DaTA-H)	0.1
*T*, duration of a period	1 (s)

**Table 2 sensors-24-05535-t002:** Performance summary in the general scenario.

Metric	DaTA	DaTA-H	STM [8]	EEM [10]	ECm [16]
Energy Consumption	lowest	higher	highest	lower	lower
Sum-Throughput	lowest	lower	highest	higher	higher
Energy Efficiency	best	worse	worst	better	better
Failed Decoded Signals	0	0	0	worst	worst

## Data Availability

The data presented in this study are available on request from the corresponding author. The data are not publicly available due to legal restrictions.

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
