# Peer review of "Energy Consumption Minimization with SNR Constraint for Wireless Powered Communication Networks"

_sensors, 2024, doi:10.3390/s24175535_

Round 1
Reviewer 1 Report
Comments and Suggestions for Authors
In this paper, the energy consumption minimization problem for the wireless powered communication networks (WPCNs) is addressed. The problem is formulated as a nonlinear programming model and a heuristic method is proposed in this study to provide some near-optimal solution. The authors provide simulation based experimental results to show better performances of their proposed technique in terms of energy consumption and efficiency compared to the related works in this study. The paper is well-organized and written well. However, several non-trivial technical issues and certain typographical/grammatical mistakes are found based on which the following suggestions are made to further improve the paper.
1. Abbreviated terms like DT-SWIFT, HAP, etc., must be expanded on their first use in abstract.
2. While discussing the other related works, it would be better to use the phrase “the researchers/authors in [ref no.]” instead of directly using the ref. no.
3. In the fourth paragraph of Introduction, authors need to provide the definition of achieved throughput for the wireless stations, PP and AP to make the STM and EEM problems more understandable to the readers.
4. As the authors have proposed an adapted version of the existing SWIFT scheme, they should point out the disadvantages of applying SWIFT scheme in the WPCNs and then state how their proposed technique overcome it in the relevant part of Introduction to express the novelty of their proposed technique.
5. A proper definition of channel gain coefficient is needed and its relation to the energy consumption and data transmission should be stated in the subsection 2.1, to clarify the significance of using it in such context.
6. In Section 3, while describing DT-SWIFT assisted time allocation method, ‘si’ is used to denote some wireless station, not a communication link or path. Since channel capacity refers to the maximum rate at which data can be transmitted over a communication path or channel, the term ‘the channel capacity of si’ mentioned at line 299 should be rectified.
7. In the subsection 4.1, some justification for formulating αi according to equation 19 along with the value of αn should be provided.
8. The mathematical formulae or procedure to calculate the network sum throughput and network energy consumption should be provided in Section 5.
9. The placement of HAP and other wireless stations in both the general and special scenarios should be pictorially shown for better understanding the differences between them.
10. Sections 3, 4 and 5 should be thoroughly checked to correct the typographical/grammatical mistakes.
Reviewer 2 Report
Comments and Suggestions for Authors
This work addresses the energy consumption minimization problem in wireless powered communication networks (WPCNs). The goal is to minimize the network energy consumption with the aid of the proposed DT-SWIPT-assisted time allocation scheme, named DaTA (using the SQP-based algorithm). Additionally, this work deals with the SNR (Signal-to-Noise Ratio) constraint to guarantee the use of a reliable communication channel for efficient data transmissions.
In many WPCN applications, such as Wireless Sensor Networks (WSNs), the Internet of Things (IoT), Wireless Body Area Networks (WBANs), etc., there is a strong requirement for real-time response (QoS), especially for critical applications where critical and non-critical network traffic coexist. Therefore, it would be useful to estimate more precisely how the proposed mechanisms for energy consumption minimization (and network sum-throughput maximization) affect the real-time response. This should take into account the specific requirements of critical applications, where QoS should be combined with the maximization of the wireless network lifetime through the minimization of energy consumption of the wireless stations. QoS in communication networks depends mostly on dynamic resource allocation and adaptive communication protocols (in this work, the TDMA scheme is referenced: “The network follows the HTT protocol [8] and adopts the TDMA protocol for access control…)
Additionally, mobility is a characteristic/requirement in many application areas for real-world scenarios. In this work's simulation model, the network's wireless nodes are fixed in specific locations: “…The coordinate of the HAP is (0, 0)… 10 wireless stations are randomly deployed within 4 to 10 meters from the HAP… in the special scenario, the 10 wireless stations are arranged in the same quadrant…” Is it possible to estimate how the mobility of the network nodes could affect the performance of the proposed scheme in terms of energy consumption (and probably QoS), taking into account that “…If wireless stations are deployed closely, the DaTA-H scheme can perform quite well...”?
In paragraph 2.2, it is mentioned: “…Before a period starts, the HAP will first allocate its own transmission time (τ0), the transmission time of si (τi), and the proportion of time that si uses to transmit information (αi)…”, where ti and ai are derived from Eqs (18) and (19) respectively. Would it be useful to have these parameters selected dynamically?
Finally, it is unclear what the exact value(s) of the channel traffic which is(are) used in the simulation scenario (it’s only mentioned that “the throughput demand, Rthi, is 0.8 bps”… and “…the results are the average of 30 sets of data(?)…”).
Comments on the Quality of English Languageminor editing is required
Reviewer 3 Report
Comments and Suggestions for Authors
The paper deals with the problem of power consumption minimization in wireless power communication networks (WPCN) and proposes a time allocation scheme using DT-SWIPT, which the authors call DaTA. The article is based on a theory verified by computer simulations; the experiment is missing. In addition to formal comments such as:
- Some abbreviations are broken down only after use
- "article" should be used instead of "paper",
I have the following questions/comments:
- Line 131: "...optimization problem may take much time to solve..." How much? Do the authors have any experience or examples of the calculation time?
- Figure 1 - How the behavior of the system will change during the direct connection (IT) of individual S stations (as is possible, e.g., in a 5G network)
- Line 468: "...10 wireless stations are randomly deployed within 4 to 10 meters..." What is the exact distribution of these stations? I suppose the geometric distribution (possible overlaps, etc...) of the stations matters a lot - does it?
- Chap. 5.3, or equation (25) – is this a generally accepted definition of efficiency in WPCN? If it were efficiency, as it is generally known, it is appropriate to requalify the state so that the resulting efficiency is lower than 1 (100%). According to the formula, the efficiency is higher with more elements in a given network. Perhaps it would be appropriate to rename the evaluated quantity without the term "efficiency". Based on the above, it is complicated to read the values/curves in Fig. 10 (higher value is better?). What if there are a few more elements in a given network? Will the network be even more efficient?)
I recommend evaluating efficiency as the ratio of the sum of usefully consumed energy to the total energy supplied to the system.
- Chap. 5.3 I apologize - I get lost in the mentioned amount of commented results and outputs - would it be possible to include an overview table in the text (based on SWOT logic or similar)?
Round 2
Reviewer 1 Report
Comments and Suggestions for Authors
Authors have provided point-to-point reply to all the raised concerns of mine and revised the manuscript accordingly.
Reviewer 2 Report
Comments and Suggestions for Authors
Overall, the authors' responses to the comments can be considered marginally sufficient.
Comments on the Quality of English LanguageMinor editing of English language required.
Reviewer 3 Report
Comments and Suggestions for Authors
First of all, I would like to thank the authors for their responsible approach. I accept all their answers/explanations and find them sufficient. Except in response to my last comment to clarify the conclusions of chapter 5.3 (and relevant subsequent chapters). In the given scope and range of the article, the reader really gets lost in the amount of data and claims; summarization (e.g. in the form of a table) would have helped the article a lot and would have gained in clarity (the authors' addition of a single sentence to the conclusions did not help this goal much). At the same time, I note that the eventual inclusion of suggested change into the next version of the article is already up to the authors’ will.
